# An Edge-Fog Architecture for Distributed 3D Reconstruction and Remote Monitoring of a Power Plant Site in the Context of 5G

**DOI:** 10.3390/s22124494

**Published:** 2022-06-14

**Authors:** Vinicius Vidal, Leonardo Honório, Milena Pinto, Mario Dantas, Maria Aguiar, Miriam Capretz

**Affiliations:** 1Department of Electrical Engineering, Federal University of Juiz de Fora, Juiz de Fora 36036-900, Brazil; vinicius.vidal@engenharia.ufjf.br (V.V.); maria.aguiar@engenharia.ufjf.br (M.A.); 2Department of Electronics Engineering, Federal Center for Technological Education of Rio de Janeiro, Rio de Janeiro 20271-110, Brazil; milena.pinto@cefet-rj.br; 3Department of Computer Science, Federal University of Juiz de Fora, Juiz de Fora 36036-900, Brazil; mario.dantas@ice.ufjf.br; 4Department of Electrical and Computer Engineering, Western University, London, ON N6A 1G8, Canada; mcapretz@uwo.ca

**Keywords:** multiple 3D scanning, edge-fog architecture, fog robotics, 5G remote monitoring

## Abstract

It is well known that power plants worldwide present access to difficult and hazardous environments, which may cause harm to on-site employees. The remote and autonomous operations in such places are currently increasing with the aid of technology improvements in communications and processing hardware. Virtual and augmented reality provide applications for crew training and remote monitoring, which also rely on 3D environment reconstruction techniques with near real-time requirements for environment inspection. Nowadays, most techniques rely on offline data processing, heavy computation algorithms, or mobile robots, which can be dangerous in confined environments. Other solutions rely on robots, edge computing, and post-processing algorithms, constraining scalability, and near real-time requirements. This work uses an edge-fog computing architecture for data and processing offload applied to a 3D reconstruction problem, where the robots are at the edge and computer nodes at the fog. The sequential processes are parallelized and layered, leading to a highly scalable approach. The architecture is analyzed against a traditional edge computing approach. Both are implemented in our scanning robots mounted in a real power plant. The 5G network application is presented along with a brief discussion on how this technology can benefit and allow the overall distributed processing. Unlike other works, we present real data for more than one proposed robot working in parallel on site, exploring hardware processing capabilities and the local Wi-Fi network characteristics. We also conclude with the required scenario for the remote monitoring to take place with a private 5G network.

## 1. Introduction

In recent years, the increasing level of automaticity has played an essential role in people’s daily lives, providing more comfortable experiences and safer and reliable tasks to humans in the industrial environment [1]. Large industrial plants are typically associated with high risks, demanding periodic and standard inspections [2,3]. Methods and technologies have been proposed for inspecting structures, primarily through point cloud generation based on sensors, such as light detection and ranging (LiDAR) and 3D cameras, or algorithms such as structure from motion (SfM), among others [4,5,6].

Different researchers demonstrated remote monitoring solutions to autonomous industrial plants [7,8,9,10]. The 3D reconstruction is an essential and challenging analytical solution for large structures in this environment [11]. For a complete scene comprehension, several views from different perspectives are needed [12]. Other challenges also arise from the gathering and processing of information from various sensors. This procedure demands synchronization, process distribution, data fusion, and communications with heavy package loads.

The current literature in 3D reconstruction shows that most of the research focuses on optimizing the reconstruction quality in a centralized manner [11,12], new reconstruction approaches [13,14,15], and enhancing algorithms performance [16]. Only a few studies have focused on developing a scalable distributed system regarding remote 3D supervision [17,18,19]. In many cases, the 3D processing application is located in the cloud. However, due to the demand for the Internet of Things (IoT)-based connected devices and applications, the cloud-based approaches may encounter sensibility for real-time systems [20,21] from the great amount of exchanged data among the devices, which demands communication bandwidth and generates high delay, energy constraints, and information redundancy [22].

Fog computing embraces the IoT concept and arises as a strategy to mitigate the burden of the traditional cloud-based approaches [21,23], improving latency, power consumption, scalability, and efficiency. The fog framework works as an autonomic system that enables the networking, storage, computing, and data management on network nodes [24,25]. In the industrial aspect, the fog paradigm can help in the processing of the data quickly and in the computation offloading to build a real-time system [26,27]. It is possible to develop a distributed 3D reconstruction system by improving, breaking down, and distributing processes through local servers communicating with the edge tier, optimizing overall performance by orchestrating data storage, processing power, and communication [20,28].

Some reasons to use an edge-fog architecture instead of traditional cloud-based approaches are that near-real-time and synchronized systems rely on communication throughput between a plethora of heterogeneous sensors, information redundancy, security, and other factors that are difficult to achieve in a cloud environment [29].

Some initial studies on the subject were previously carried out by the authors. Pinto et al. [20] detailed a mathematical simulation in order to study and analyze an edge-fog-cloud architecture in multi-robots. The authors of Silva et al. [19] presented a real case study for a 3D reconstruction approach that uses an edge-fog architecture. Different from them, this research work analyzes the cost of each node process, considering more than one real robot as an Edge Node in the architecture.

This paper proposes an improved system approach for 3D environment scanning and reconstruction, based on an edge-fog architecture focusing on large environments, especially power plants. The resultant reconstruction will be able to provide further applications, such as remote monitoring or virtual crew training.

The architecture scalability will be shown in the results by exploring more effectively the network capabilities to enhance the number of working nodes. The performance will be optimized by balancing edge and fog processing, considering the available individual computing power and the offloading between the nodes. This work also evaluates a private 5G network application in this scenario, given the practical results obtained by the current infrastructure in a real power plant environment. It presents a brief discussion on how this new technology can benefit the overall distributed processing. Therefore, the main contributions of this work can be summarized as follows:Development of a distributed 3D reconstruction and imaging framework for remote monitoring that uses an optimized, scalable edge-fog architecture for computing resources and network communication.A study and analysis of the proposed approach in a real application of an electric power plant facility.A study on the application of a 5G private network and its benefits in the proposed framework to assist in the distributed processing scalability.

The rest of the manuscript is organized as follows. Section 2 provides the background and presents an overview of related contributions. Section 3 details the implemented robot, its mathematical and construction foundations, and the proposed architecture characteristics, with process distribution evaluation. The experimental results and 5G application are discussed in Section 4. Finally, the concluding remarks and ideas for future works are in Section 5.

## 2. Background and Related Works

This section presents the current work found in the literature regarding IoT applications, focusing on network distributed processing for remote facility monitoring. It highlights the imaging, 3D reconstruction, and sensor fusion in the IoT context. Related works are also presented on how the concept of 5G and fog computing brings benefits when applied to remote applications.

It is possible to identify many solutions involving 3D reconstruction for monitoring in this new era of 5G. A proof of concept in real applications of distributed processing in an edge-fog architecture, such as the one presented in this paper, offers a significant contribution to the field with process allocation, solution scalability, and 5G benefits in a real environment.

### 2.1. Remote Monitoring of Power Facilities, 3D Reconstructions, and Fog Computing

In the application of manufacturing, the IoT technology is a powerful tool that can be used for predictive maintenance, statistical analysis, and energy optimization, among others. Industrial IoT emerges as an opportunity to enhance the traditional methods for manufacturing monitoring [30], benefiting the analysis and management of processes. The performance condition of the devices involved in an industrial facility can be remotely supervised and controlled [31], which generates a considerable amount of information from heterogeneous devices. The fog computing provides data processing and storage locally at the IoT device, with faster response services.

For a real implementation of fog computing, an appropriate architecture is necessary. Fog computing is closer to edge devices to support vertical and latency-sensitive operations. In addition, it provides scalable, layered, and distributed processing and storage in network application [32]. Several works in the literature also define the challenges, benefits, and issues related to fog computing [33,34]. We can find, for example, the necessary attention to time constraints in distributed applications, with the concept of near-real-time in fog computing, as stated in [35].

This computing paradigm usage in virtual reality (VR) applications and IoT environments is detailed in [36]. An interesting application of fog computing is presented in Sarker et al. [37]. The authors proposed an edge-fog architecture for a robotic application that relied mainly on edge-fog devices, using the fog layer for simultaneous localization and mapping (SLAM) calculations. Another edge-fog architecture application was proposed by [38], where the robot has to pick up boxes autonomously with remote data processing. Results also point to how high latency could invalidate a user interface-assisted robotics application.

Several works proposed technologies and methods to deal with the remote inspection and monitoring of power plants [39,40,41]. For instance, in Sato et al. [42], the authors developed a crawling robot for inspection of a disaster situation in a nuclear power station. The 3D reconstruction is performed by LiDAR, RGB, and RGB-D sensors, processed in an offline way through SfM 3D reconstruction.

The authors of Peng et al. [43] proposed a 3D reconstruction approach for a substation with a complete processing pipeline. The mobile LiDAR scanning robot provided information for the point cloud registration, meshing, and texture application in their approach. Unlike this work, their application relied only on offline processing without real-time application and scalability.

Another interesting approach was detailed by Guo et al. [44], based on the idea of interconnected network applications with several local sensors and servers, remote servers, and databases. Still, the substation was reproduced in a 3D CAD model, so monitoring of actual conditions and interference on-site was not performed.

It is worth mentioning that many works used fog robotics (FR) architecture for environment reconstruction and SLAM [37], 3D recognition [45], and image processing [38]. These many solutions tried to avoid processing data in the cloud [29,46]. Most of these works improved the processing time and power usage. However, they lacked application scalability and computation offloading, essential components of an FR application.

### 2.2. Edge-Fog-Cloud Applications in the IoT Context Using 5G

The work of Aleksy et al. [47] describes how new applications in industrial and robotics scenarios demand real-time response and high computational power simultaneously, often not found in devices at the edge. The 5G comes in handy to solve this problem as a local industry network and is simulated throughout the work in many applications, even involving monitoring. They prove how the response time of 5G directly affects the control in a robotic arm manipulator when processed remotely.

In Shahzadi et al. [48], the authors introduce how distributed processing in layers could take place using a 5G network. They discussed the concepts, from hardware elements to data offloading and hierarchy levels of edge, fog, and cloud computing. They conclude that fog computing brings the capabilities of the cloud closer to the IoT devices, diminishing delay issues, in a more local approach. They proposed a method to optimize throughput in the network by distributing tasks among the several layers and nodes.

Some current applications take this concept to a more specific practical scenario. The authors in [49] propose a network called X-IoCA, an internet of robotic things. From the vast heterogeneity of IoT sensors and data types, along with devices at the edge and different protocols (BLE, Wi-Fi, Lora), the final network is integrated using 5G for fast message exchange between devices and users. They apply the whole concept by coordinating ground vehicles, drones, edge devices, and user interfaces in a search and rescue mission, described thoroughly in the paper.

Mapping and 3D reconstruction are also benefited by using 5G capabilities and network layered architecture, as demonstrated in [50]. They used an edge-fog-cloud architecture to deal with LiDAR, camera reading, data, and processing offload to cope with the SLAM algorithms at the fog layer, and finally, the cloud capabilities to perform the dynamic mapping operations. They described the system and processes that each layer is composed of and presented the system’s performance due to the number of vehicles working as Edge Nodes. Similarly, Shin et al. [51] presented an edge-fog-cloud processing architecture to deal with object recognition along roads, describing how each layer is built and planned to deal with expected big data from heterogeneous sources (cameras, LiDAR, GPS, among others). Results are presented related to system recognition performance and scalability of Edge Nodes, which is a crucial goal in these application scenarios.

## 3. Materials and Methods

This section describes the necessary hardware to perform the monitoring and 3D reconstruction tasks. It gives the mathematical and algorithm formulation required to achieve this goal. We also introduce the proposed architecture with the methodology on how to split and organize the process in an edge-fog scenario. From this point on, the term “point cloud” refers to a collection of 3D points that together represent an entity’s shape in 3D virtual space and should not be confused with “cloud”, which refers to cloud computing.

### 3.1. Robot Hardware

A robot was built to cope with the 3D and imaging requirements proposed in this work (Figure 1), provided with network access capabilities for distributed processing. It is composed of a Livox Horizon LiDAR scanner with built-in inertial measurement unit (IMU), a C925s USB full HD camera, two Dynamixel AX-18A servo motors, a 12.0 V and 8400 mAh battery pack, and a wireless Gbyte router with a local network. It is controlled by an NVidia Jetson Nano board.

The camera was calibrated to give the point cloud accurate color according to the LiDAR sensor’s reference frame. The calibration process was inspired by the Github repository provided by the manufacturer [52]. The servo motors were responsible for pan and tilt movements in a range of 360 and 120 degrees, respectively. The built-in IMU was used to measure roll and pitch angles and compensate for ground imperfections, with the measurements submitted to a low-pass filter in the driver. The pan angle measurement came from the servo motor, with a resolution of 0.088 degrees/tick.

The robot can communicate through the local Wi-Fi network to a laptop computer posed as the Fog Node. All the nodes use Robot Operating System (ROS) Melodic on Ubuntu 18.04 as a middleware to deal with synchronization.

### 3.2. Robot Control and Algorithms

In order to create the robot’s path, it is necessary to set the pan and tilt angles, with a predefined step in degrees, covering the space first in the tilt and then in the pan directions. This is performed by using a nonlinear control algorithm [53]. The robot covers a total of $n_{p}\cdot n_{t}$ waypoints for data acquisition considering $n_{p}$ pan and $n_{t}$ tilt views. Angles and axes to control our robot’s rotation and odometry are presented in the diagram of Figure 2, where ϕ, θ, and ψ represent the roll, tilt, and pan angles around Z, X, and Y axes, respectively.

Equation (Equation 101) computes the rotation transformation from the LiDAR to the pan point of view (PPV) frames ${}_{l}R^{ppv}\in {\mathrm{SO}}^{3x3}$ (the group of special orthogonal matrices), considering ϕ and θ as the current acquisition odometry. The point cloud captured by the LiDAR is $C_{l}$. It is therefore possible to transform every point $P_{l_{k}}\in R^{3}$ in $C_{l}$ according to Equation (Equation 102), to create $C_{{ppv}_{i}}$.
(1a)$${}_{l}R^{ppv}=\left[\begin{array}{ccc} cos\phi & -\cos\theta \cdot \sin\phi & \sin\theta \cdot \sin\phi \\ sin\phi & \cos\phi \cdot \cos\theta & -\cos\phi \cdot \sin\theta \\ 0 & \sin\theta & \cos\theta \end{array}\right]$$
(1b)$$C_{{ppv}_{i}}=⋃\left({}_{l}R^{ppv}\cdot P_{l_{k}}\right),\;\;\forall P_{l_{k}}\in C_{l}$$

The readings are accumulated by stacking them in sequence, forming point cloud $C_{ppv}$ in the PPV frame with $n_{t}$ tilt angles according to Equation (Equation 2).
(2)$$C_{ppv}=⋃C_{{ppv}_{i}},\;\;i\in \left[1,n_{t}\right]$$

In order to represent color, every point $P_{{ppv}_{k}}\in C_{ppv}$ (in this case, in homogeneous coordinates, so $P_{{ppv}_{k}}\in R^{4}$) is projected in every captured image $I_{t}:\Omega^{t}\subset Z^{2}\to {\left[0,2^{8}\right]}^{3};\left(u,v\right)\mapsto I_{t}\left(u,v\right)$ gathered for this PPV. Equation (Equation 303,Equation 302,Equation 305) presents the use of extrinsic matrix ${}_{ppv}T^{cam}\in R^{4x4}$ (Equation (Equation 303)) and intrinsic $K_{cam}\in R^{3x3}$ (Equation (Equation 302)) to obtain the final pixel value $p_{h_{k}}\in Z^{3}$, in homogeneous coordinates (Equation (Equation 305)). The group of images is called $I_{v}=\cup I_{t},\forall t\in \mathrm{PPV}$, and should contain $n_{t}$ images by the end of the tilt travel in every PPV.
(3a)$${}_{ppv}T^{cam}=\left[\begin{array}{cc} R_{cam_{3x3}} & t_{cam_{3x1}}\\ 0 & 1 \end{array}\right]$$
(3b)$$K_{cam}=\left[\begin{array}{ccc} f_{x} & \alpha_{y} & c_{x}\\ \alpha_{x} & f_{y} & c_{y}\\ 0 & 0 & 1 \end{array}\right]$$
(3c)$$p_{h_{k}}=K_{cam}\cdot {}_{ppv}T^{cam}\cdot P_{{ppv}_{k}},\forall P_{{ppv}_{k}}\in C_{ppv}$$
where $R_{cam}\in {\mathrm{SO}}^{3x3}$ and $t_{cam}\in R^{3x1}$ are extrinsic rotation and translation extrinsic components for the camera, respectively; *f* and *c* are the camera’s focus and optical center coordinates; and α represents skew and distortion along in both axes.

At the end, to obtain the final pixel coordinates $p_{k}\in Z^{2}$, and therefore the point’s color, one must divide the homogeneous $p_{h_{k}}={\left[\begin{array}{ccc} u & v & w \end{array}\right]}^{T}$ by its last coordinate, so $p_{k}={\left[\begin{array}{cc} u/w & v/w \end{array}\right]}^{T}$.

When the robot finishes a PPV travel, the current pan angle ψ is used to transform point cloud $C_{ppv}$ from the PPV to the local frames, creating $C_{lo}$. Equation (Equation 401) defines how ${}_{ppv}T^{lo}\in R^{4x4}$ is calculated based on ψ and the LiDAR mounting offsets $t_{lc}={\left[\begin{array}{ccc} x_{lc} & y_{lc} & z_{lc} \end{array}\right]}^{T}$. With that said, $C_{lo}$ is calculated as in Equation (Equation 402), in a similar fashion to Equation (Equation 305).
(4a)$${}_{ppv}T^{lo}=\left[\begin{array}{cccc} \cos\psi & 0 & \sin\psi & x_{lc}\\ 0 & 1 & 0 & y_{lc}\\ -\sin\psi & 0 & \cos\psi & z_{lc}\\ 0 & 0 & 0 & 1 \end{array}\right]$$
(4b)$$C_{lo}=⋃\left({}_{ppv}T^{lo}\cdot P_{{ppv}_{k}}\right),\;\;\forall P_{{ppv}_{k}}\in C_{ppv}$$

The current $C_{lo}$ is compared to the last one, acquired from the previous PPV, defined as $B_{lo}$, to avoid duplicate readings in the same region. A Kd-tree search process [54] is applied for every $P_{{lo}_{k}}\in C_{lo}$, looking for neighbors in $B_{lo}$. $P_{{lo}_{k}}$ is removed from $C_{lo}$ if more than a threshold number of neighbors is found, namely, $Th_{nn}\in Z$, in a region of radius $r_{nn}\in R$.

With a reduced $C_{lo}$, point normals are estimated according to [55]. Statistical outlier removal filter is applied following the algorithm presented in [56].

Finally, according to Equation (Equation 5), registration occurs by stacking the remaining points in $C_{lo}$ onto the accumulated point cloud Acc.
(5)$$Acc=Acc⋃C_{lo}$$

### 3.3. Proposed Architecture

As stated by the survey presented in [57], there are different approaches and understandings of edge-fog computing. Architectures are characterized depending on how the computing and communications capabilities are distributed [58].

Some applications, such as low-level robotics solutions, have the main process running at the edge processors and devices. In such cases, data acquisition and critical processes require real-time performance, so the communication latency to send, process, and return the output cannot rely on most network environments or cloud-based servers. The higher hierarchical levels integrate the results for the user.

On the other hand, data analysis and computationally intensive applications typically use an architecture that connects the end device to fog or even cloud servers. The end device forwards the acquired data, and all processing is executed in dedicated hardware, as reported in [28,59].

The architecture proposed for this application exploits the benefits of using edge and fog tiers, avoiding high delays and lack of synchronism that appear when placing processes in a cloud layer. It is illustrated in Figure 3. In remote distributed 3D reconstruction, low-level filters and high-level accumulating processes clearly separate data and information; raw high-resolution images and point clouds can be filtered in the individual embedded nodes, whereas asynchronous stacking can be carried out in more centralized, powerful hardware. The proposed approach breaks down the problem into different layers considering the amount of data, computing power, sensor proximity, and network throughput [19].

The amount of information passing through the network decreases by preprocessing the raw data from each individual node, reducing the required bandwidth. In the proposed near-real-time remote 3D reconstruction system, each robot is an Edge Node, and controls its own camera and LiDAR, producing 240,000 points per second and 1920 × 1080 RGB resolution images at 10 Hz, processing and forwarding the results. The final computing-heavy, asynchronous, and demanding processing is executed in the Fog Node. This approach configures the edge-fog computing-based IoT architecture [60].

### 3.4. Distributed Processing Methodology

Every relevant process from the formulation presented in Section 3.2 was evaluated regarding execution time *per unit* (PU), input memory size (IMS), and output memory size (OMS). The time measurement unit chosen was the time requested for the CPU to perform 100 for loops with a summation of two integer values. Data were gathered repeatedly in different inspection scenarios for approximately two seconds and processed in sequence. Table 1 presents the average results for PU, IMS, and OMS. The laptop computer conceived as the Fog Node contains a 9th generation i7 processor with 12 model 9750-H@2.60 GHz cores, and presented a PU average of 72 nanoseconds. For the Jetson Nano, the same time was measured as 95 nanoseconds.

The processes are divided between the edge and fog tiers in each respective node, taking into account mainly sensor proximity, computing power requirements, and network bandwidth consumption.

The data from Table 1 suggests that reducing the image dimensions and encoding reduces the data in the network drastically since it avoids sending the full HD image process (3rd row, in italic). The full-resolution image is only mandatory when coloring the point cloud for a more precise environment reproduction, not achievable with the low-resolution image. The PU required to color the point cloud is considered irrelevant compared to the time necessary to acquire data. That being said, it is worth filtering and coloring the point cloud at the edge to benefit the network traffic at the expense of processing a fraction of the total PU from Table 1. The entire amount of data being transmitted in each message would go from 11.310 MB (OMS for raw data in the first row) to 5.808 MB (OMS for fourth and fifth row combined) between edge and Fog Nodes.

Figure 4 presents the workflow illustration, with the numbers from Table 1 respectively placed. The Edge Node is mainly responsible for acquiring data and implementing basic data filtering and fusion, while the Fog Node receives data and deals with high-level computing effort, final storage demands, and user interaction.

One should notice that the data size in the last process is accumulated whenever a new point cloud message arrives, so the traffic to storage and user interface is constantly increasing.

## 4. Results in Experimental Environments

### 4.1. Improvements from the Edge-Fog Architecture When Compared to Edge-Based Approach

This section presents the benefits of using an edge-fog architecture for data and processing offload instead of trusting all the workflow to an edge device, also known as edge computing [61]. The authors have applied this technique previously in a controlled scenario [19]. In this work, the proposed architecture was tested in a real case of a power plant environment, where Figure 1b was captured.

When all the processes from both edge and Fog Nodes in Figure 4 are conceived inside the Edge Node, the image is sent in the same way as in our edge-fog architecture, but the constantly increasing accumulated point cloud is repeatedly transferred after it is finished, straight to storage at the Fog Node and to the user interface. That leads to increased latency and throughput requirements in the network, and data are not constantly transmitted, which is not efficient in itself. This fact is presented with a throughput graph for both architectures’ scenarios in Figure 5. For the sake of simplicity, only one minute of acquisition is presented. Note that the network potential is more deeply explored in the edge-fog architecture, since the edge-based architecture concentrates much more time for data transferring after accumulating everything from a PPV travel. The red curves show that data are mainly acquired until approximately 26 seconds. The data transfer takes about 11 seconds, as we see a higher value in the red curves, saturating the network bandwidth in our experimental setup. On the other hand, the Edge Node is constantly offloading data and processing for every waypoint in the edge-fog architecture scenario. The blue curves present this behavior by constantly oscillating between sending only the image (lower throughput) and adding the point cloud on top of it (higher throughput).

We measured CPU and RAM behavior for architecture evaluation in each device by using monitoring tools available in Ubuntu 18.04. Each running process is analyzed individually inside each node. The CPU usage relates to the average of all cores working in parallel. RAM is measured by summing the value used by each process in a sampling instant, with the maximum value representing peak RAM. Both CPU and RAM usage are compared to the total available capacity in each node, leading to percentage results.

Table 2 presents the results on CPU usage and peak RAM for both architectures. Unlike [37], the results are focused on both edge and fog devices and the difference in behavior for both architectures’ processing requirements. Since in the edge-based architecture the computer works only as a user interface, only the edge device data is presented.

Table 2 shows a much more balanced scenario in terms of average CPU activity during acquisition, with a 28.2% reduction in the proposed architecture. A significant result was also obtained for peak RAM demand at the edge, which was reduced from 88.3% to only 41.8% of total capacity. It relates to the data offloading goal and enables more sensors to be added to the Edge Node in future works, e.g., GPS.

### 4.2. Application Scalability

As a next step from our previously mentioned work [20], this section presents results for real case scenarios with up to five robots working in parallel as Edge Nodes, communicating with a central processing Fog Node. The entire application is supported and synchronized by the ROS framework, which has the Fog Node as the master device. Messages timestamps are based on the master device clock, with millisecond order precision.

Table 3 gives the evolution in RAM usage and CPU activity in the Fog Node for each number of parallel robots scanning the environment, which tend to evolve linearly with the number of parallel processes demanded by each robot.

From the linear approximation of CPU activity, it is the first variable to become the bottleneck in the application. In our model, the Fog Node would be facing a 99.56% usage when processing the work of nine Edge Nodes. Nonetheless, the fog tier is composed of just one node in the current experiment, which could be expanded if more Edge Nodes were added to the solution.

Figure 6a presents the throughput curve for one-minute acquisition. Only the curves for one, three, and five robots are plotted to simplify the information and present the overall behavior visually. Figure 6b shows the average values plus standard deviation. It is possible to notice the increase in network demand and saturation at about 8 MiBps due to hardware limitations on bandwidth, which may further impact latency and generate packet loss.

Consider a simple linear polynomial fit to the average and standard deviation in the throughput evolution graph of Figure 6b. We can assume that seven robots will already require approximately 8.30 MiBps plus 0.55 MiBps deviation, which will be considered as network bandwidth saturation with our Wi-Fi infrastructure. From our practical experience, this many robots are not enough to cover the rooms existent in a power plant. Applying more robots will also lead to high latency and loss in near-real-time values from the saturated network.

As a final qualitative remark, we present both scanned sites point clouds and images in Figure 7 and Figure 8, representing the machine and turbine rooms, respectively. The red circles indicate where the robots were positioned for scanning.

### 4.3. Relevance of 5G Network for the Monitoring Scenario

Our current proposed hardware uses a local Wi-Fi network for communication between the nodes. The results have confirmed that this technology is limited in terms of bandwidth and throughput. The network capabilities become the bottleneck in terms of scalability as the number of parallel robots increases. As presented in [61], 4G network could be a solution with a larger signal range, but still lacks mainly in bandwidth, downlink, and uplink resources to serve an entire power plant in the current context of remote monitoring involving imaging and 3D reconstruction. Therefore, 5G arises as the best suit for this application.

Many works are investigating 5G by means of simulation, as already shown in Section 2. Still, several references point to real networks being applied in industrial scenarios, describing the network capabilities. Our work is mainly interested in the downlink and uplink capabilities of private 5G networks since it will avoid the current bottleneck issue of network saturation due to low bandwidth.

The work of [62] evaluates a commercial 5G base station network behavior when communicating to on-board devices in drones flying at different heights. They witnessed values of up to 742 Mbps and 46 Mbps for downlink and uplink rates, respectively. Based on this effort, ref. [61] presents a discussion and methodology on process allocation in a drone visual–inertial navigation problem based on network characteristics, among others. It simulates downlink and uplink rates ranging from 320–6400 Mbps and 40–800 Mbps, respectively. Finally, in a more similar environment compared to ours, ref. [63] presents the application and tests of a commercial 5G base station in a substation in Brazil, aiming to perform a proof of concept for digital twin and grid automation applications. For a laptop, the average values for downlink and uplink rates were recorded as 958 Mbps and 83 Mbps, respectively.

Based on these values gathered from real data, the following predictions are assumed for the application of a 5G private network in the power plant remote monitoring scenario:From the data gathered and shown in Figure 6, as a conservative approach, each robot is expected to require 5.67 MiBps of throughput, considering average plus one standard deviation, which is equivalent to 47.56 Mbps.Each Fog Node will be saturated from processing data of nine Edge Nodes due to the CPU utilization constraint.Nine parallel Edge Nodes will demand a throughput of 9 × 47.56 Mbps = 428.04 Mbps in each Fog Node connection.The value of 83 Mbps for uplink rate recorded in [63] is already enough for each Edge Node to send data according to our experimental first requirement, so any greater values will only help in network robustness and latency decrease.

The result for the number of fog and Edge Nodes allowed to be working simultaneously is evaluated according to the network downlink capability, with the result demonstrated in Figure 9. The results are plotted for downlink values ranging from 700–6400 Mbps. The way we estimate it is simply adding an Edge Node once the necessary bandwidth is available, and a Fog Node when the number of Edge Nodes requires a new one. We consider an average of nine robots per room (data estimated locally to cover the main view spots, still allowing the Fog Node to be physically closer to the edge tier). The growing number of robots will allow all the main rooms at the power plant to be monitored from a downlink rate of approximately 3000 Mbps or higher (seven rooms).

## 5. Conclusions and Future Work

This research has proposed an edge-fog architecture as a solution to support a methodology for distributed 3D reconstruction for near real-time remote supervision. Experimental scenarios using our developed robot have demonstrated that the system proposed in this study provides an efficient approach compared to processing everything in an edge device, while also providing scalability and reducing network demands that would exist if raw data were transmitted to the Fog Node. The applied methodology successfully uses a combination of local processing and network resources to deal with challenging computing requirements. This offloading concept can be beneficial for many applications, such as UAV mapping, SLAM methods in general, or other types of inspection and autonomous vehicles. Similar approaches could also be used for different data types, such as image processing, given proper adaptions.

The work exploits the benefits of edge and fog tiers for balancing the whole solution, avoiding the use of cloud computing, and resulting in a more secure and local application. Any application involving the cloud, e.g., a user interface, will receive filtered and secured data from our developed system and is meant for future works.

The proposed architecture has been applied to a multi-robot scenario with up to five robots. Special attention was given to network and Fog Node behavior. The Fog Node CPU activity and peak RAM presented an approximately linear increasing pattern. The data lead to expectations on the network scalability and design, which is a remarkable goal in an industrial IoT scenario.

The data were submitted to a scalability evaluation in a local/private 5G network scenario to be implemented at the power plant, given the proper parameters found in the literature. The results prove the tremendous increase in dealing with a growing number of Edge Nodes due to the network’s higher bandwidth and throughput characteristics. The expansion in the architecture proved itself feasible, given the proper growth in the number of Fog Nodes and their location in the power plant.

Future work will evaluate the position of the nodes in more rooms at the power station and assess the impact on the network behavior. With a private 5G network implementation, new data can be acquired, and tests will be performed for reliable remote monitoring. Optimization will also be performed on process allocation in the presence of the 5G network and nodes’ position concerning the antennas. 

## Figures and Tables

**Figure 1 sensors-22-04494-f001:**
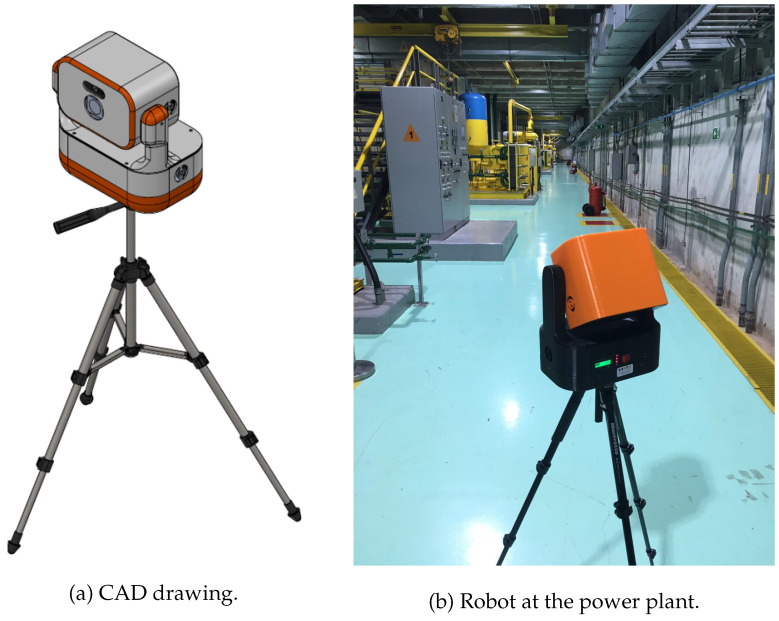
Implemented robot. (**a**) CAD drawing. (**b**) Robot at the power plant.

**Figure 2 sensors-22-04494-f002:**
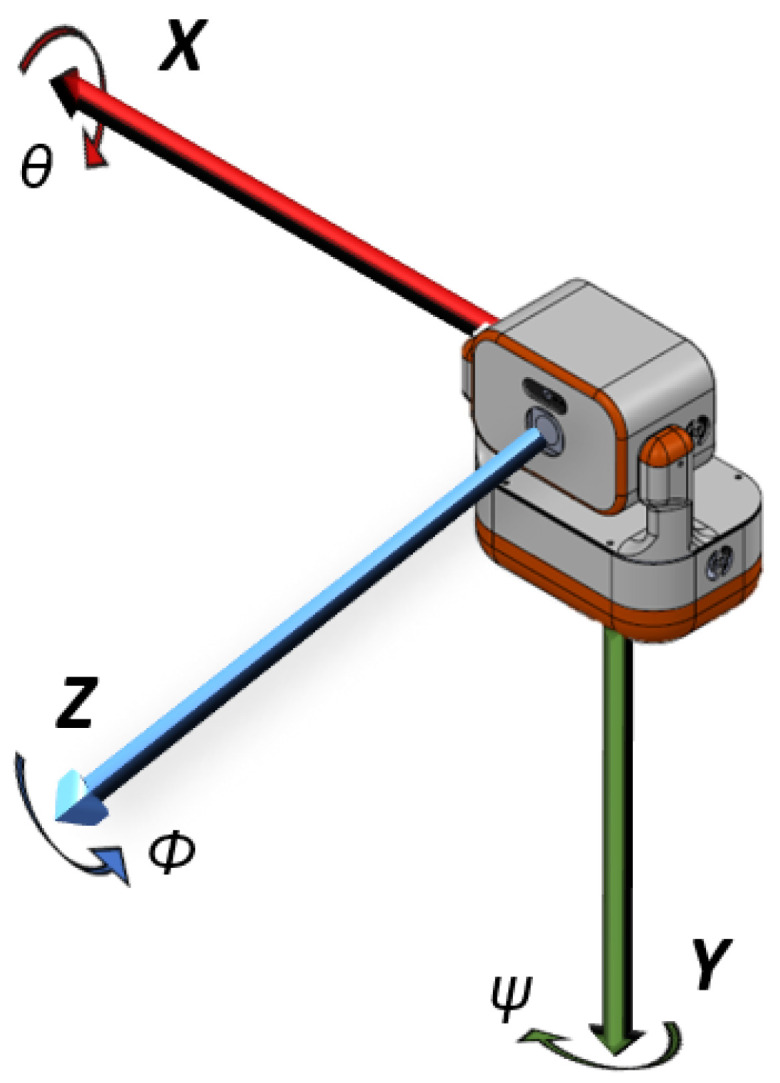
Robot’s diagram illustrating the axes and their respective rotation angles.

**Figure 3 sensors-22-04494-f003:**
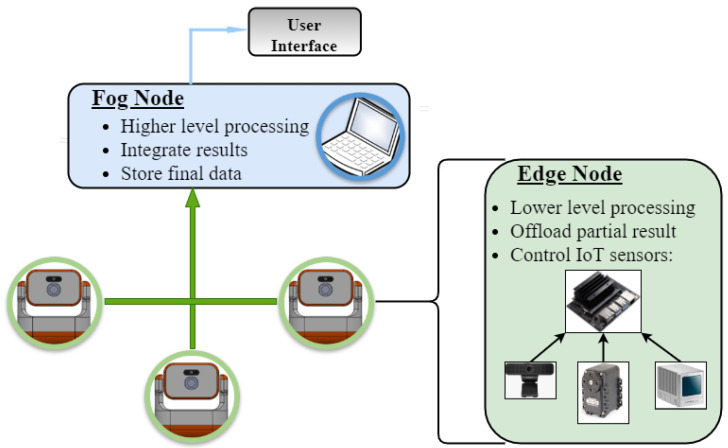
Edge-fog architecture illustration, with a summary of each node’s part in the architecture.

**Figure 4 sensors-22-04494-f004:**
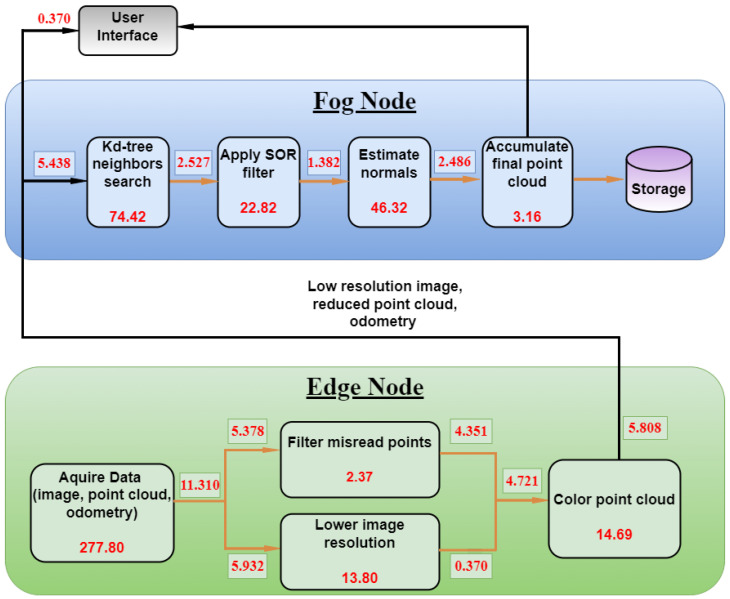
Logical processing workflow split into edge and Fog Nodes. Numbers inside the boxes indicate the processing PU ($\times 10^{5}$). Numbers in the arrows indicate the average message sizes in MB. All data should refer to Table 1.

**Figure 5 sensors-22-04494-f005:**
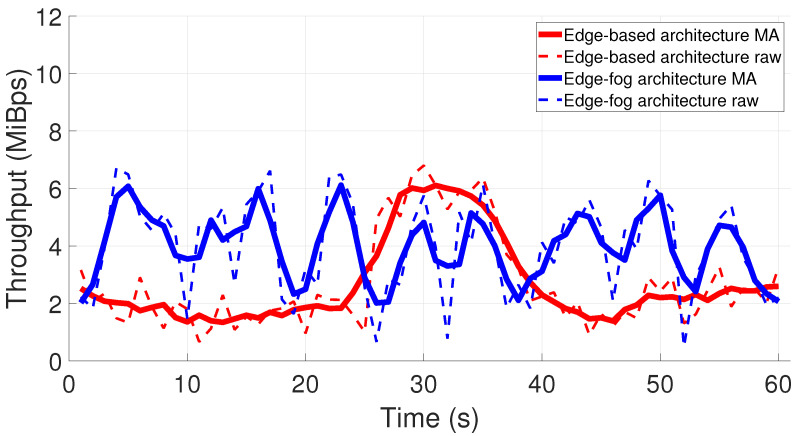
Throughput for both architectures during one minute of scanning. Dashed lines represent raw data, and solid lines represent the moving average.

**Figure 6 sensors-22-04494-f006:**
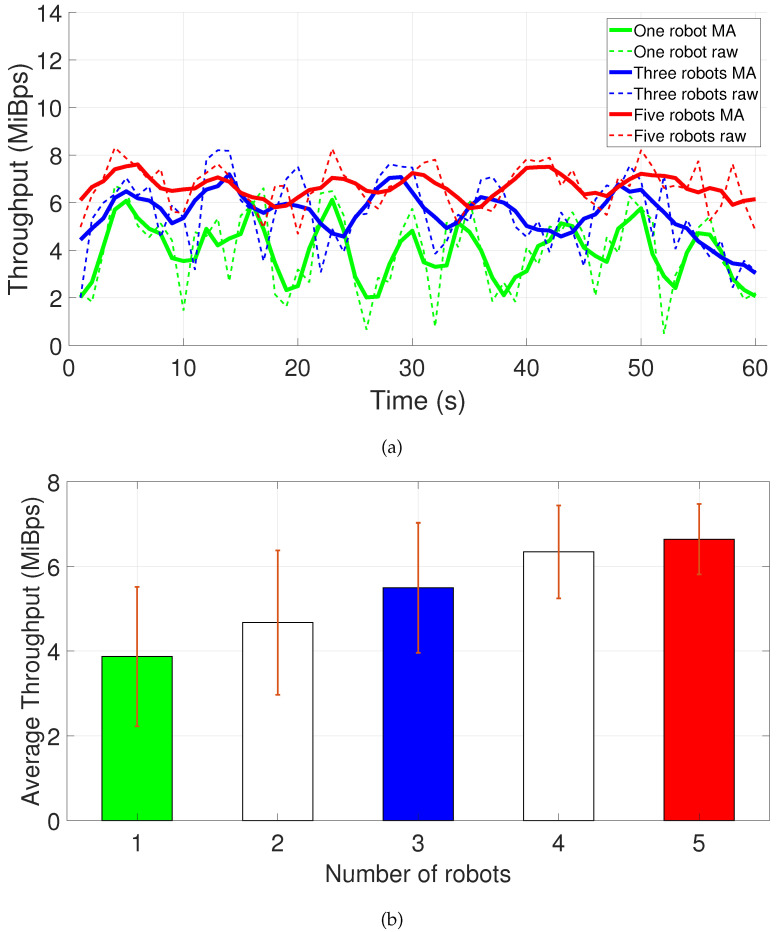
Throughput evolution for an up to five parallel robots scenario: (**a**) One minute throughput curve. (**b**) Average rate plus standard deviation. Dashed lines stand for raw data, while solid lines represent the moving average.

**Figure 7 sensors-22-04494-f007:**
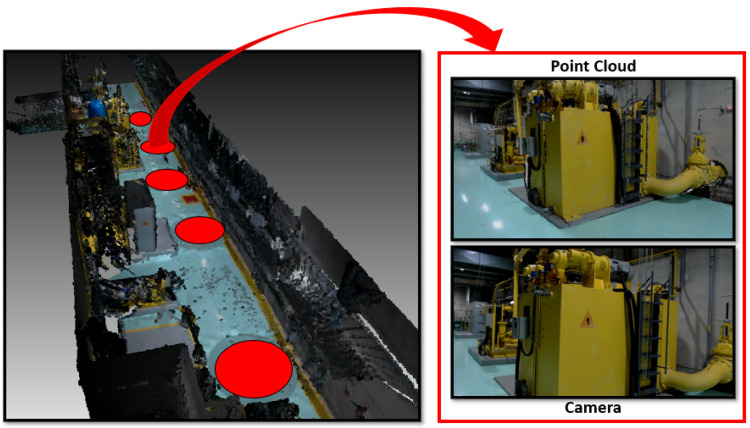
Machine room, scanned by 5 robots. The azimuth view is presented with ceiling removed for better interpretation. Red dots indicate the robots positions. Point cloud and camera views are illustrated.

**Figure 8 sensors-22-04494-f008:**
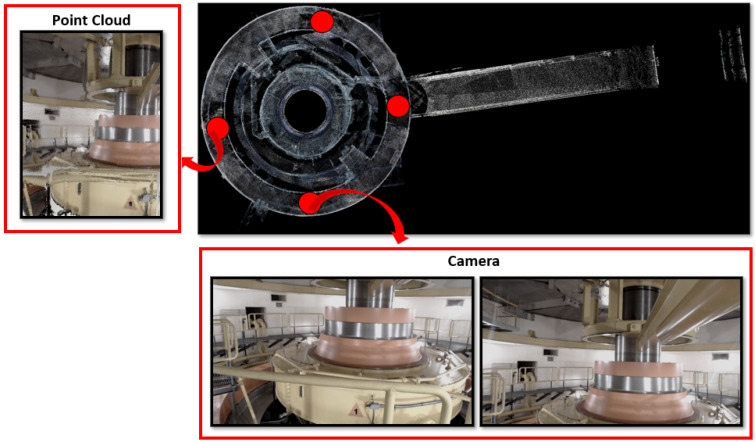
Turbine room, scanned by 4 robots. The top view is presented, with the red dots indicating the robots positions. Point cloud and camera views are illustrated.

**Figure 9 sensors-22-04494-f009:**
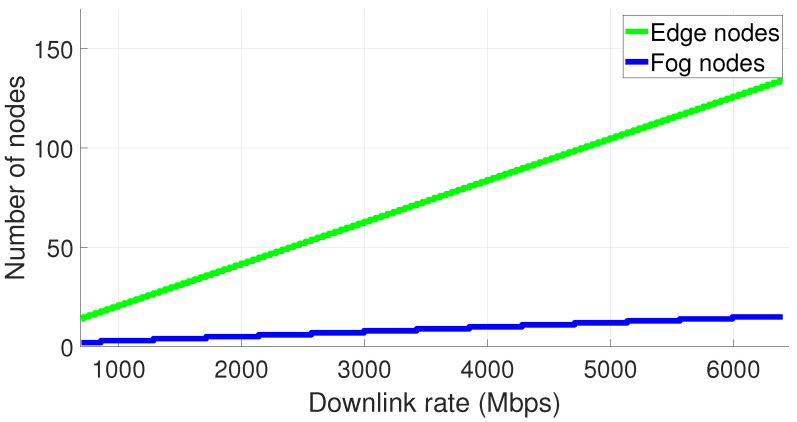
Evolution of the number of edge and Fog Nodes according to the downlink rate availability from the private 5G network.

**Table 1 sensors-22-04494-t001:** Average results for PU, IMS, and OMS for every relevant processing step. The process in italic (3rd row) is avoided by lowering image resolution.

Process Name	PU $\left(\times 10^{5}\right)$	IMS (MB)	OMS (MB)
Acquire data	277.80	11.310	11.310
Filter misread points	2.37	5.378	4.351
*Send full HD image*	*412.71*	*5.932*	*5.932*
Lower image resolution	13.80	5.932	0.370
Color point cloud	14.69	4.721	5.438
Kd-tree neighbors search	74.42	5.438	2.527
statistical outlier removal (SOR) Filter	22.82	2.527	1.382
Normal estimation	46.32	1.382	2.486
Accumulate final point cloud	3.16	2.486	-

**Table 2 sensors-22-04494-t002:** CPU activity and peak RAM results for both edge-based and edge-fog scenarios.

	Edge-Based	Edge-Fog	
	Edge	Edge	Fog
**CPU activity (%)**	91.4	63.2	30.2
**RAM (%)**	88.3	41.8	5.9

**Table 3 sensors-22-04494-t003:** CPU activity and peak RAM in the Fog Node for up to five parallel working robots.

	Number of Robots				
	1	2	3	4	5
CPU activity(%)	30.2	33.6	42.9	53.3	63.7
RAM(%)	5.9	11.0	16.2	19.9	27.1

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
