# Peer review of "An Edge-Fog Architecture for Distributed 3D Reconstruction and Remote Monitoring of a Power Plant Site in the Context of 5G"

_sensors, 2022, doi:10.3390/s22124494_

Round 1
Reviewer 1 Report
Improve the quality of figures.
Reviewer 2 Report
The paper address an interesting topic which combines several up-to-date technologies (AR, Edge, and Fog computing, 3D reconstruction, remote monitoring and sensing, 5G), which are used combined to give an efficient remote monitoring system for a power plant.
However, for this paper to be accepted for publications, the following points should be addressed:
- The paper needs a careful proof reading and language improvement. The below are just examples for weak/wrong sentences or expressions: (In line 10: for data and processing and offload), line 240 (to and understanding),
- Formatting issues: line 245, line 291 )from the Table, (put the table number)
Technical issues:
- Please provide more elaboration on the “point cloud” concept/definition
- I understood the reason for choosing 5G network, but can you please add a paragraph discussing how the performance of the proposed system is affected/changed when using other networks (e.g. WiFi or 4G) ? This is important since the 5G mentioning appears on the papers’ title.
- In the paper, several angles are used (theta, phi, psi, ..), can you please create a three dimensional diagram that have these angles.
- In line 206, can you please elaborate more on the symbol O^3x3
- Please put a reference for the kd-tree search algorithm mention in line 231
- The authors chose the usages of edge, fog. However, nothing is mentioned about the cloud layer? Will it be necessary to have the cloud in the proposed system? If not, can you please provide a discussion on that regard (i.e. the merit behinds the network architecture and main components (Fog, edge and cloud).
- The authors mentioned the potential usage of 5 robotics in parallel (line 343). However, nothing is mentioned about who these robots will coordinate the operation between themselves? Do they need to have a mechanism to synchronize their operation?
Reviewer 3 Report
-there are many typos, proofreading is required to polish the paper. Example, in abstract 'architecture for data and processing and offload'
-please add the link of the repository: "The calibration process was inspired by the Github repository provided by the manufacturer".
-in table 2, what is the point of adding column 'user interface' if both values are empty, remove it or fill it.
-how did you measure CPU and RAM activities, are the presented values the average of how many trails? more details on this are required.
- you can add the following missing related works:
[1] Ivanov, Donat. "Fog robotics distributed computing in a monitoring task." Computer Science On-line Conference. Springer, Cham, 2020.
[2] Cai, Xiaofeng, et al. "Robot and its living space: A roadmap for robot development based on the view of living space." Digital Communications and Networks (2020).
Reviewer 4 Report
This is a well-motivated work and a technically very strong paper perfectly illustrating the benefits of the 5G technology in an Industry 4.0 setting.
